# Pain and Cortisol in Patients with Fibromyalgia: Systematic Review and Meta-Analysis

**DOI:** 10.3390/diagnostics10110922

**Published:** 2020-11-09

**Authors:** Edurne Úbeda-D’Ocasar, Victor Jiménez Díaz-Benito, Gracia María Gallego-Sendarrubias, Juan Antonio Valera-Calero, Ángel Vicario-Merino, Juan Pablo Hervás-Pérez

**Affiliations:** 1Department of Physiotherapy, Faculty of Health, Camilo José Cela University, Villanueva de la Cañada, 28692 Madrid, Spain; eubeda@ucjc.edu (E.Ú.-D.); gmgallego@ucjc.edu (G.M.G.-S.); javalera@ucjc.edu (J.A.V.-C.); 2Department of Sport Sciences, Faculty of Health, Camilo José Cela University, Villanueva de la Cañada, 28692 Madrid, Spain; vjdiaz-benito@ucjc.edu; 3Department of Nursing, Faculty of Health, Camilo José Cela University, Villanueva de la Cañada, 28692 Madrid, Spain; avicario@ucjc.edu

**Keywords:** fibromyalgia, chronic pain, cortisol, treatment, meta-analysis

## Abstract

This systematic review and meta-analysis sought to gain further insight into the relationship between cortisol reactivity and chronic widespread pain in patients with fibromyalgia. The studies selected were those conducted in adults with fibromyalgia that were random controlled, non-controlled or observational. Studies were excluded if they examined diseases other than fibromyalgia or if they did not report on pain or cortisol. Twelve studies met inclusion criteria. Data were extracted into tabular format using predefined data fields by two reviewers and assessed for accuracy by a third reviewer. The methodological quality of the studies was assessed using the PEDro scale. Data Synthesis: Of 263 studies identified, 12 were selected for our review and 10 were finally included as their methodological quality was good. In the meta-analysis, we calculated effect sizes of interventions on pain indicators and cortisol levels in patients with fibromyalgia. A small overall effect of all the interventions was observed on pain tolerance and pressure pain thresholds, yet this effect lacked significance (ES = 0.150; 95%CI 0.932–1.550; *p* > 0.05). Conclusions: While some effects of individual nonpharmacological therapeutic interventions were observed on both cortisol levels and measures of pain, our results suggest much further work is needed to elucidate the true relationship between chronic widespread pain and cortisol levels in patients with fibromyalgia.

## 1. Introduction

Fibromyalgia (FM) is a complex multifactorial syndrome characterized by chronic widespread pain that is often accompanied by fatigue, cognitive problems and sleep disturbances causing a considerable decline in patient quality of life [1,2,3]. Over the past decades, several criteria have been defined for the classification, diagnosis and detection of FM, but criteria reflecting the present understanding of this disease that could help clinicians and researchers are lacking. This lack is clinically relevant during the clinical practice since criteria have to be valid, reliable and consistent to identify, assess and classify patients with FM and to make the most accurate treatment decision. According to research criteria, patients are required to have pain in the axial skeleton, above and below the waist and on both sides of the body. With the goal of a multifaceted diagnosis, in 2013, a working group on fibromyalgia was created to assess new diagnostic approaches to help identify FM in clinical practice. This diagnostic taxonomy (ACTTION-APS Pain Taxonomy, AAPT) classifies chronic pain according to the dimensions: (1) core diagnostic criteria, (2) common features, (3) common medical comorbidities, (4) neurobiological, psychosocial and functional consequences, and (5) putative neurobiological and psychosocial mechanisms, risk factors, and protective factors [4,5].

Current research efforts suggest that the underlying basis of the symptoms of FM could be the altered physiology of the central nervous system, whereby the abnormal processing of pain signals plays an important role in its pathogenesis. This nociceptive system dysregulation may arise from a combination of interactions among the autonomic nervous system, neurotransmitters, cytokines and hormones, among others [6].

Cortisol is an essential steroid hormone produced in the suprarenal cortex within the suprarenal gland [7,8,9]. Levels of cortisol both in blood and saliva vary throughout the day, reaching their peak approximately at 8 am and showing lowest levels between 12 pm and 4 am, or 3 to 5 h after the onset of sleep [8]. Cortisol concentrations are related to stress levels and blood glucose concentrations.

The hypothalamic–pituitary–adrenal axis (HPA) is considered a mediator of cortisol production. HPA activation has been associated with the severity of chronic musculoskeletal pain [7,10,11,12] and with fluctuations in perceived pain [13]. However, when analyzing the effects of cortisol on chronic pain, clinical studies have detected their inverse relationship. Thus, some investigations have shown that a higher cortisol concentration is related to a lower pain intensity [14,15], and accordingly, lower cortisol concentrations have been linked to greater levels of pain [7,16]. Pain symptoms are among the five main reasons patients with FM visit the emergency room [17].

Several explanations have been put forward for the low cortisol reactivity observed in patients with FM. For example, a possible reason for HPA hypofunction in FM would be the low secretion of corticotropin releasing hormone (CRH) by the hypothalamus and secondary atrophy of the suprarenal glands due to this low chronic stimulation because of reduced adrenocorticotropic hormone (ACTH) levels [18]. In contrast, it has been also proposed that the cause could be a reduced response of suprarenal cortisol to ACTH. Hence, diminished adrenocortical sensitivity to ACTH could be attributed to the complete regulation of suprarenal receptors, but genetic variation and morphological changes in the suprarenal gland along with atrophy or diminished volume could also contribute [19,20].

Due to the need for more evidence when trying to understand the relationship existing between cortisol concentrations and chronic widespread pain, this systematic review and meta-analysis sought to analyze and summarize the latest literature examining this topic in patients with FM. Main goals were: (a) to identify studies conducted in the past 10 years analyzing variations in cortisol levels and perceived pain produced in response to a treatment intervention or between subjects with and without FM; (b) to assess the methodological quality of the studies identified; (c) to calculate the effect sizes of the interventions proposed on cortisol and pain; and (d) compare the effectiveness of the interventions.

## 2. Materials and Methods

### 2.1. Data Acquisition, Search Strategy and Eligibility

Our review was conducted according to international PRISMA (Appendix A) guidelines for systematic reviews and meta-analyses [21,22]. An electronic search was conducted between October and December 2019. Potentially relevant studies were identified by searching the following databases; Pubmed-MEDLINE; ScienceDirect; and PEDro.

The publication cutoff was December 2019. For the Medline search, the sensitive strategy proposed by the Cochrane Collaboration was used [23]. The search terms were obtained from MeSH through PubMed and included: cortisol, fibromyalgia and chronic fatigue, fibromyalgia syndrome and pain as follows: (cortisol OR “cortisol level”) AND (fibromyalgia OR “chronic fatigue syndrome” OR “chronic fatigue fibromyalgia syndrome”) AND pain. The review was limited to articles published in English and Spanish, and a thorough search of grey literature was not undertaken. The reference list of each included study and relevant reviews were examined for potential studies. The studies selected were those conducted in adults with FM that were random controlled, non-controlled or observational. Studies were excluded if they examined diseases other than fibromyalgia or if they did not report on pain or cortisol or not exist data for estimate effect size. The study selection process for this review is illustrated in Figure 1.

### 2.2. Methodological Quality

The quality of the studies described in the articles selected (external and internal validity, statistics reports) was assess using the PEDro scale which has been validated for this purpose [24,25]. This variable was independently assessed by three of the authors (JPH-P, VJD-B, EU-D). Any disagreement was solved by consensus.

### 2.3. Statistical Analysis

All data were recorded in an Excel spreadsheet (v. 2016) for Windows including outcome variables, sample size and the remaining information in each original article. In cases where the authors did not report the size of the effect, effect sizes of interventions were calculated on the outcome measures cortisol and pain through the software Effect Size determination Program [26]. The data compiled were the means, standard deviations (SD) and the contrast statistics provided in the original articles (Student *t*, Mann–Whitney U and Snedecor F). Forest plots were constructed using Graph Pad Prism V. 8.3.1 (San Diego, California, USA). The positive bias caused by the standard deviation for small samples was treated through the corrected d value [27]. A fixed effect model was employed to determine the homogeneity of the treatment measures of the original studies. Homogeneity was determined according to the procedure developed by Hedges and Olkin through the Q statistic, distributed according to Chi-square with k-1 degrees of freedom, where k is the number of effect sizes [27]. For each study, we provide the corrected effect size along with corresponding 95% confidence intervals (95%CI). In the original reports using non-parametric tests, we considered the η2 provided according to the proposal of Cohen. Intervals were defined as small (≤0.06), medium (0.06–0.14) or large (≥0.14). For original studies using parametric tests, effect size was considered small when the statistics value was a score around 0.20; medium when around 0.50 and large when around 0.80, as proposed by Cohen [28].

## 3. Results

### 3.1. Data Synthesis

From the 263 records initially identified in this review over the past 10 years, 251 records were excluded (115 were duplicates; 30 were letters to the editor, notes, abstracts or meeting presentations; 7 articles were not written in English or Spanish; 79 articles were excluded since the main issue addressed was not related to fibromyalgia, cortisol and pain; and 22 because pain and cortisol outcomes were not provided). Finally, 12 articles fulfilling the inclusion and exclusion criteria were selected (see Figure 1).

### 3.2. Methodological Quality

Each of the 12 studies identified were scored according to the criteria of the PEDro scale (see Table 1). This scale categorizes studies as “good” methodological quality and low bias risk when awarded a score of 5–10 [29]. This score was observed in all but two studies, which were designated a score of 3. These were thus excluded from further analysis. The remaining 10 articles were awarded scores from 5 to 8. The mean score was 5.60 ± 1.07 (Table 1).

### 3.3. Data Extraction and Analysis

The full texts of the 10 studies finally selected, 18, 30–36, 38, and 40, were independently examined by three of the authors (EU-D, VJD-B and JPH-P) to extract the information: participants and demographic data, study objectives, intervention protocol, outcome measures and results obtained (Table 2).

Of the 10 studies reviewed, four were non-randomized controlled [30,33,35,36], two randomized controlled [18,34], two observational [31,32], one case-control [38] and one pilot study [40].

Participants of the 10 studies reviewed were 382 subjects (all women), of whom 212 were allocated to an FM group (all diagnosed with FM) and 170 to a control group. Two studies lacked a control group [32,36]. Mean ages were 50.03 ± 5.36 years in the FM groups and 48.41 ± 7.17 years in the control groups. Body mass index (BMI) values provided by all except three of the studies [18,31,40] were 26.44 ± 1.45 and 26.44 ± 0.89 for patients and controls, respectively.

### 3.4. Combined Effects of Outcome Measures

For valid estimates of effect size for the meta-analysis model, the predictive variables compared were cortisol, pain tolerance threshold (PTT) and pressure pain threshold (PPT) (Figure 2 and Figure 3).

Figure 2 shows descriptive statistics and effect sizes found in the 8 studies in which effects on cortisol were examined. Among these studies, significant medium effect sizes were detected in the studies of Pegado et al. [31] and Stehlit et al. [38] (ES = 0.017; 95%CI 0.066–1.421; *p* < 0.05; and ES = 0.79; 95%CI 0.189–1.365; *p* < 0.05, respectively). Significant large effect sizes were observed in the studies by Genc et al. [18], Pernambuco et al. [34], and Torgrimson-Ojerio et al. [40] (ES = 0.807; 95%CI 0.230–1.384; *p* < 0.05; η2 = 1.630; 95%CI 0.944–2.316; *p* < 0.05; η2 = 0.74; 95%CI 0.418–1.055; *p* < 0.05, respectively). Collectively, these data indicate an overall effect of the interventions on cortisol. However, the effect of the interventions on cortisol levels was ruled out as zero was included in the confidence interval and it proved to be not significant (ES = 0.066, 95%CI 0.452–0.712; *p* > 0.05).

Figure 3 shows the studies analyzed provided according to the descriptive statistics and the effect sizes found for the changes produced in PTT and PPT. Large significant effects were observed on PTT and PPT in the studies by Pegado et al. [31] (η2 = 0.679, 95%CI 0.005–1.352) and Stehlit et al. [38] (*p* < 0.05). Significant large effect sizes were also observed in the interventions by Geiss et al.33, Genc et al. [18], Riva et al. [35], and Torgrimson-Ojerio et al. [40] (ES = 2.47; 95%CI 1.509–3.433; η2 = 2.367; 95%CI 1.590–3.144; *p* < 0.05; ES = 1.026; 95%CI 0.478–1.574; *p* < 0.05; ES = 1.329; 95%CI 0.713–1.945; *p* < 0.05). Again, a small overall effect of all interventions was observed, yet this effect lacked significance (ES = 0.150; 95%CI 0.932–1.550; *p* > 0.05).

## 4. Discussion

This review and meta-analysis sought to summarize the latest scientific literature regarding the possible relationship between cortisol levels and the pain symptoms of fibromyalgia. The results of our meta-analysis indicate some individual effects of therapeutic interventions on both cortisol levels and several measures of pain though overall effect sizes were insignificant.

Fibromyalgia affects the 2.1% of the world population and 2.4% of persons in Spain. It is diagnosed mainly in women, the reported ratio women:men varying from 2:1 [41] or 3:1 [42] to 10:1 [20,43,44]. This much larger proportion of affected women is well reflected in this review in which all study participants were women. In effect, it is difficult to find studies including both sexes and those that have done so have examined a smaller proportion of men [45,46].

The best therapeutic approach to fibromyalgia is integrating pharmacological and non-pharmacological treatments (exercise therapy, patient education and cognitive behavioral therapy) while actively involving patients in their own care process. In particular, the important role of stress reduction, sleep and physical exercise as basic self-management strategies should be stressed [1]. Pharmacological agents include analgesics, antidepressants, anticonvulsants and muscle relaxants [2]. Only one of the studies analyzed the impact of pharmacological therapy (low dose dexamethasone) in patients with FM. In response to the drug, cortisol levels were found to increase following measurement of the pressure pain threshold, and post hoc analysis of measures revealed a parallel increase in levels of pain. Individualized treatments prescribed by multidisciplinary teams including clinicians with expertise in patient education and mental health, physical or occupational therapists offer improved outcomes over pharmacological treatments alone [1].

The treatment interventions tested in the studies reviewed were often exercise related [18,32,40]. However, the therapeutic value of these exercise interventions for FM emerged as low. This may be mainly attributed to incomplete descriptions of the exercise programs tested and poor patient adherence [47]. Physical therapy was the second most common intervention tested [30,36] and one study was based on lifestyle interventions [34]. Three of the studies involved an observational assessment [31,35,38]. The reports selected for this review were required to fulfil strict inclusion criteria related to pain and cortisol and thus do not really reflect usual treatment interventions. According to Basavakumar et al., treatments most often used for FM are interventions on lifestyle, followed by medication and non-pharmacological treatments, such as physical therapy and physical exercise, along with the use of nutrition supplements [48].

In the studies selected for this review, cortisol levels were determined using diagnostic tests on blood, saliva or urine samples. Several authors have established that morning cortisol levels in serum [49], saliva [50,51] or hair [51] are lower in subjects with chronic musculoskeletal pain [52], FM [49,50] or chronic fatigue syndrome [49].

Several independent studies for this review were selected according to descriptive statistics and effect sizes reported for the effects of several predictive variables on cortisol levels. In two of these studies, significant medium size effects were observed [31,38], while in three, effects were both significant and large [18,34,40].

In a meta-analysis, Tak et al. compared 85 cases of FM and controls. These authors detected a significant reduction in baseline cortisol in all female patients with FM compared to healthy control subjects and described a participating role of HPA in functional somatic disorders including fibromyalgia [53]. Low baseline cortisol levels in FM patients were also reported in the articles reviewed here [18,30,36,40].

There is evidence of disassociation between total and free cortisol levels in patients with FM, who generally show normal free cortisol levels in plasma and saliva despite total cortisol levels being diminished [6]. Other authors have observed that salivary daily cortisol levels are reduced while cumulative cortisol levels in hair remain normal [54]. A comparison between FM patients with neck and shoulder pain and healthy controls revealed significantly lower awakening cortisol levels in the patients [55]. The disparate results obtained in the different studies reviewed are consistent with findings in the literature.

Analysis of pain in the different studies has revealed poor agreement between the high therapeutic value of exercise and adherence to exercise recommendations.

The results of the studies selected for review indicated large significant effects of several interventions on the pain tolerance threshold and pressure pain threshold [18,31,33,35,38,40].

In general, the effects observed were variable. This variability could be explained by the heterogeneity of both interventions and study designs. Cortisol measurement protocols (in urine or blood) and pain tolerance or threshold tests were conducted using different instruments and protocols. This disparity might explain why Torgrimson-Ojerio et al. [40] noted significant reductions in cortisol in their FM group while others report significant reductions in the control group [34]. Variation in pain was described in the study by Geiss et al., who used the pressure pain threshold method [33], while Pernambuco et al., Riva et al., and Stehlik et al., measured pain more subjectively though the fibromyalgia impact questionnaire (FIQ) and visual analogue scale (VAS), respectively [34,35,38]. While this variability suggests these data should be analyzed with caution, they seem to indicate that a guided combined physical therapy/aerobic exercise program involving sessions two days a week over 5 weeks could help relieve pain and reduce plasma cortisol levels [18,30]. Long-term experimental studies based on multifaceted programs could help standardize interventions and detect larger more consistent pain alleviating effects of exercise.

In response to a treatment intervention such as those described in the studies reviewed here, patients with FM show improved pain tolerance and perceived pain thresholds and these improvements are largely reflected by a better perceived health state [1] However, although the results of the studies examined here and those of most studies in the literature are promising, sample sizes have been small. Thus, larger therapeutic interventions are needed to support the evidence available before their generalized implementation. While fibromyalgia is much better understood and managed today than before, more work is needed on non-pharmacological approaches to symptom treatment to further improve patient quality of life.

A limitation of this review is that some studies could not be included in our meta-analysis because they either did not report on cortisol and pain or were published in a language other than English or Spanish. There was also some heterogeneity detected, which indicated variation in the degree of association between the intervention tested and effects on pain and cortisol. The low number of studies means we were unable to provide a good estimate of the overall effect on cortisol and pain. One of the possible reasons for this is that adherence to treatment in patients with FM is not always the most adequate due to the specific characteristics in relation to pain in these patients, determining that only a limited number of randomized clinical trials have addressed this issue, and the total number of patients in those studies. Our search revealed the scarcity of research on this topic.

## 5. Conclusions

The low number of clinical studies identified precludes establishing any clear relationship between cortisol levels and perceived pain when examining the effectiveness of several therapeutic interventions. The studies reviewed were of medium methodological quality, thus revealing a need for higher-quality trials conducted on larger patient numbers designed to examine variations in cortisol levels in response to pain in patients with fibromyalgia. The large number of different study designs identified in the literature led to similarly different effects observed. This variation, along with the different treatment interventions, could explain while overall effect sizes in relation to correlation between pain and cortisol did not prove significant. The scarcity of randomized controlled trials on this topic determines a need for much further work to elucidate the true relationship between chronic widespread pain and cortisol levels in patients with FM.

## Figures and Tables

**Figure 1 diagnostics-10-00922-f001:**
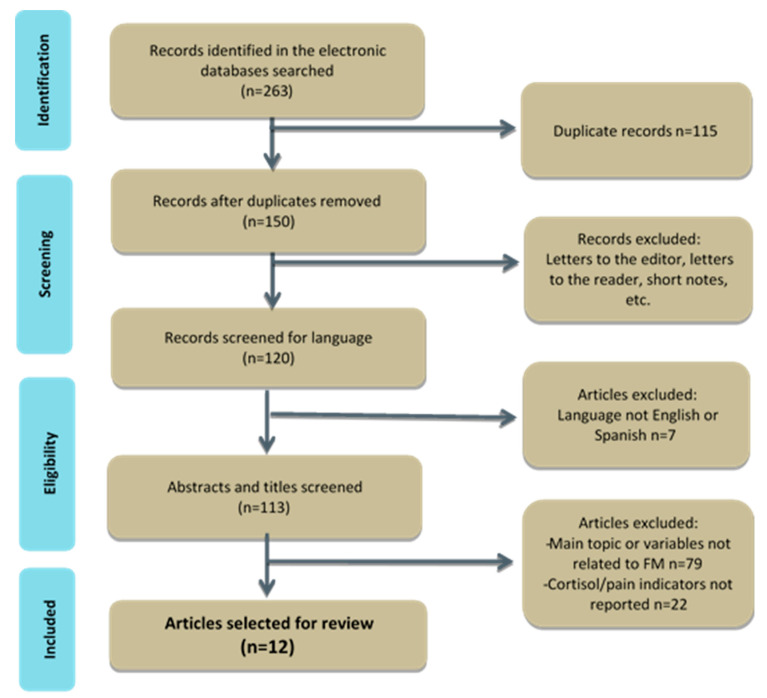
Articles identified for this review. FM = Fibromyalgia.

**Figure 2 diagnostics-10-00922-f002:**
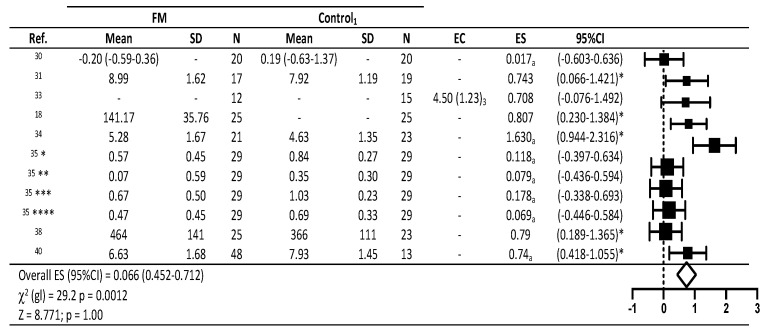
Descriptive statistics and forest plot of effects of fibromyalgia treatment interventions on cortisol levels. CS = contrast statistic; 1: comparison group, other treatment applied; 2: median (min-max) percentage change; 3: F (gl): Snedecor contrast F statistic (degrees of freedom); a: η2: Eta squared; 5: all measures; η2 as proposed by Cohen: <0.06 small, ≥0.06 to <0.14 medium and ≥0.14 large [28]; * = *p* < 0.05 in the original study analyzed; ES: effect sizes as proposed by Cohen: <0.2 small, ≥0.5 to <0.8 medium and ≥0.8 large28. 35 * = Sample 1: 35 ** = Sample 2; 35 *** = Sample 3: 35 **** = Sample 4 (Salivary cortisol samples during the late afternoon, before and after dinner, in the evening, and at bedtime).

**Figure 3 diagnostics-10-00922-f003:**
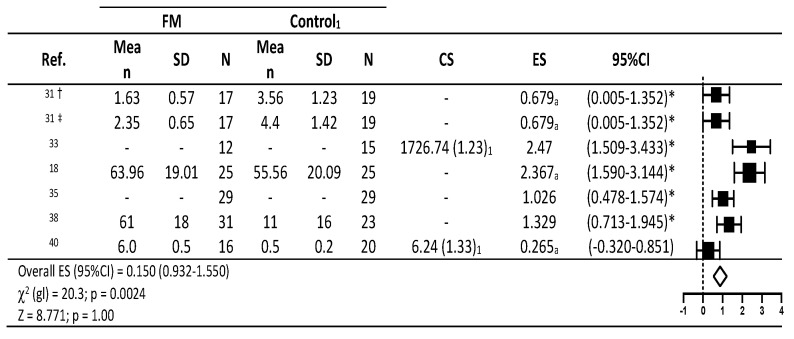
Descriptive statistics and forest plot of effects of fibromyalgia treatment interventions on pain tolerance threshold and pressure pain threshold. CS = contrast statistic; 1: F (gl): Snedecor contrast F statistic (degrees of freedom); a: η2: Eta squared; †: PPT pressure pain threshold (in reference 31); η2 as proposed by Cohen: <0.06 small, ≥0.06 to <0.14 medium and ≥0.14 large28; * *p* < 0.05 in the original study analyzed; ES: effect sizes as proposed by Cohen: <0.2 small, ≥0.5 to <0.8 medium and ≥0.8 large [28]. ‡: PTT pain tolerance threshold (in reference 31); η2 as proposed by Cohen: <0.06 small, ≥0.06 to <0.14 medium and ≥0.14 large28; * *p* < 0.05 in the original study analyzed; ES: effect sizes as proposed by Cohen: <0.2 small, ≥0.5 to <0.8 medium and ≥0.8 large [28].

**Table 1 diagnostics-10-00922-t001:** PEDro scores of the studies identified.

Reference	Study Type	PEDro											
		S	2	3	4	5	6	7	8	9	10	11	TOTAL
Genc et al., 2015 [18]	Randomized controlled	+	+	+	+	-	-	-	+	+	+	+	7
Alp et al., 2014 [30]	Non-randomized controlled	+	-	-	+	+	-	-	+	+	+	+	6
de Abreu et al., 2012 [31]	Observational cross-sectional	+	-	-	+	-	-	-	+	+	+	+	5
Garrido et al., 2017 [32]	Observation longitudinal	+	-	-	+	-	-	-	+	+	+	+	5
Geis et al., 2012 [33]	Non-randomized controlled	+	-	-	+	-	-	-	+	+	+	+	5
Pernambuco et al., 2018 [34]	Clinical randomized controlled	+	+	+	+	-	-	+	+	+	+	+	8
Riva et al., 2010 [35]	Non-randomized controlled	+	-	-	+	-	-	-	+	+	+	+	5
de Oliveira et al., 2018 [36]	Non-randomized controlled	+	-	-	+	-	-	-	+	+	+	+	5
Schertzinger et al., 2018 [37]	Longitudinal	+	-	-	-	-	-	-	-	+	+	+	3
Stehlik et al., 2018 [38]	Case-control	+	-	-	+	-	-	-	+	+	+	+	5
Tanwar et al., 2018 [39]	Non-randomized controlled	+	-	-	-	-	-	-	+	-	+	+	3
Torgrimson-Ojerio et al., 2014 [40]	Pilot	+	-	-	-	-	+	-	+	+	+	+	5

S: selection criteria; 2: random allocation; 3: concealed allocation; 4: similarity at baseline; 5: subject blinding; 6: therapist blinding; 7: assessor blinding; 8: >85% measures for initial participants; 9: intention to treat; 10: between-group statistical comparisons; 11: point and variability measures. None of the selected articles had a conflict of interest.

**Table 2 diagnostics-10-00922-t002:** Characteristics and results of the studies reviewed.

Ref.	Demographics	Objectives	Intervention	Outcome Measures	Results (*p* Values)
[35]	FM: *n* = 29 52.1 ± 8.9 years BMI 27.1 ± 5.9 Control: *n* = 29 52.7 ± 8.4 years BMI 25.0 ± 3.5	To determine salivary cortisol and pain levels in different conditions such as stress, upon awakening, 30 min, 60 min after awakening, etc.	Collection of saliva eight times at the time points: arrival at the hospital (4:45 pm), late afternoon, late evening, CAR and after leaving the hospital the next day (5:05 pm)	-Salivary cortisol -Pain (VAS)	Patients in FM showed declining cortisol levels over the day, most pronounced in the morning (CAR). Cortisol levels lower in FM versus Control. Differences significant between groups in 1st, 2nd, 5th and 8th measurement (*p* = 0.010, *p* = 0.035, *p* = 0.001 and *p* = 0.050 respectively) Difference between groups in pain perception
[31]	FM: *n* = 17 53 ± 7.98 (42–69 years) Control: *n* = 19 53.32 ± 6.46 years	To examine changes produced in cortisol and their correlation with pain, depression and quality of life in postmenopausal women with fibromyalgia	Blood cortisol measured after 8 h of sleep along with pain threshold and pain tolerance	-Cortisol levels measured over three alternate days -Algometry post blood collection on 18 Tender Points (TPs) (kg/cm^2^) until participant reports “starting to feel pain” and “can’t take it anymore”	Pain threshold *p* > 0.0001 between groups Pain tolerance *p* > 0.0001 between groups α = 5% Significant difference between groups No link between cortisol levels and pain threshold or tolerance
[33]	FM: *n* = 12 50 ± 2.07 years BMI 26.30 ± 0.363 Control: *n* = 15 41 ± 2.98 years BMI 26.47 ± 1.01	To determine cortisol and IL-6 responses after measuring PPT at TPs	4-day study: measurements at baseline and after low dose overnight dexamethasone (0.5 mg)	-Blood cortisol -PPT -TP count	Cortisol levels significantly increased post PPT measurement *p* < 0.04 PPT measurement led to higher pain levels when measures were analyzed post hoc
[30]	FM: *n* = 19 51 (25–64 years) BMI 27.1 Control: *n* = 20 48 (36–54 years) BMI 26.5	To assess the effects of balneotherapy on the hypothalamic-pituitary axis	3-week balneotherapy program consisting 20-min sessions 5 days/week	-Blood cortisol (g/dL) Pre/Post intervention -TP count	Cortisol levels fell (*p* = 0.002) after the Week 3 balneotherapy session compared to baseline levels on Day 1 in controls. Cortisol levels rose by 19% in the Week 3 session compared to Day 1 in the FM group (*p* = 0.005), and fell by 20% in the control group. TP count fell by 7% (*p* = 0.02) in FM
[40]	FM: *n* = 20 52.0 ± 1.4 years Control: *n* = 16 4852.2 ± 1.5 years	To determine cortisol levels associated with pain following an exhaustive exercise test	Fasting treadmill exercise to V0_2_ peak of some 20 min duration	-Blood cortisol -PTT	PPT reduced in FM (*p* = 0.001) No significant differences in cortisol pre and post exercise (*p* = 0.10)
[18]	FM1: *n* = 25 36.9 years FM2: *n* = 25 35.1 years	To assess pain and blood cortisol levels following a home stretching and aerobic exercise program	6 weeks of home exercise or 6 weeks home + aerobic exercise	-Blood cortisol -PTT -TP count	Pain reductions observed in FMI (*p* < 0.025) and FM 2 (*p* < 0.001) TP count reduced Significant differences in cortisol levels from baseline to first and second measurement (*p* = 0.014) Post-hoc contrast statistic increased from 1st to 2nd measurement
[32]	FM: *n* = 14 51.07 ± 12.38 years BMI 23.65 ± 4.00 TPs 17.10 ± 0.05	To examine the effects of functional respiratory training on pain and their correlation with cortisol levels	Diaphragm breathing exercise intervention. Measurements made over 12 weeks: first 4 weeks control followed by 8 weeks of exercise.	-Sleep quality -Algometry pain tolerance threshold (PTT) -Urine cortisol	Significant increases produced in PTT between week 4 and 12 (*p* < 0.05) in occiput, low cervical and 2nd rib No changes in cortisol levels during intervention
[34]	FM: *n* = 21 51.43 ± 11.26 years BMI 26.51 ± 5.18 Control: *n* = 23 48.26 ± 11.03 years BMI 26.82 ± 4.22	To assess salivary cortisol, pain (FIQ) and TPs	Health Education program	-Salivary cortisol -Pain (FIQ) -TP count	Cortisol levels rose in FM (*p* = 0.02), but not in Control. Pain scores improved in FM (*p* < 0.02) but remained the same in Control
[36]	FM: *n* = 24 45.9 ± 2.89 years BMI 26.04 ± 2.52	To determine salivary cortisol levels and pain after a 3-month Swedish massage program	Massage program = 24 × 40 min sessions (2 afternoon sessions/week)	-Salivary cortisol -Pain (FIQ)	No significant differences before and after the 3- month intervention. Pre- post intervention differences significant for first session and first month (*p* < 0.001), but significance lost after 2nd and 3rd month
[38]	FM: *n* = 31 57 ± 8 years BMI 28.4 ± 5.7 Control: *n* = 23 57 ± 10 years BMI 27.4 ± 5.4	To correlate chronic pain with morning blood cortisol levels and leg pain	Comparative study	-Blood cortisol -Pain (VAS)	Differences significant between groups in cortisol levels (*p* = 0.01) and pain perception (*p* < 0.001)

BMI: body mass index; FM: fibromyalgia; TP: tender points; VAS: visual analogic scale; FIQ: fibromyalgia impact questionnaire; PTT: pain tolerance threshold; PPT: pressure pain threshold.

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
