# Peer review of "Pain and Cortisol in Patients with Fibromyalgia: Systematic Review and Meta-Analysis"

_diagnostics, 2020, doi:10.3390/diagnostics10110922_

Round 1

Reviewer 1 Report

The systematic review and meta-analysis of cortisol reactivity and wide-spread pain in fibromyalgia patients by Úbeda-D’Ocasar et al., including random controlled, non-controlled and observational studies strength is the lack of an updated complete systematic review on the topic which could shed light on this controversial association. Weaknesses are the low number of studies included (10 out of 269, after filtering for quality, and finally only 8 for effect size analysis), and their heterogeneity. Not surprisingly leading to a lack of significance on the analyzed data.

According to Methods and Figure 1. it is not clear why the randomized clinical trial published Jan 2019, cite 51 was not included in this study. Additional studies on the topic: pain and cortisol levels on FM trials are available in the 10-year period indicated by the authors. For example, the following two, among several other unmentioned:

Pereira Pernambuco A, de Souza Cota Carvalho L, Pereira Leite Schetino L, Cunha Polese J, de Souza Viana R, d' Ávila Reis D. Effects of a health education program on cytokines and cortisol levels in fibromyalgia patients: a randomized controlled trial. Adv Rheumatol. 2018 Aug 3;58(1):21. doi: 10.1186/s42358-018-0022-z. PMID: 30657084.

Roman P, Estévez AF, Miras A, Sánchez-Labraca N, Cañadas F, Vivas AB, Cardona D. A Pilot Randomized Controlled Trial to Explore Cognitive and Emotional Effects of Probiotics in Fibromyalgia. Sci Rep. 2018 Jul 19;8(1):10965. doi: 10.1038/s41598-018-29388-5. PMID: 30026567; PMCID: PMC6053373.

Giving the impression that the review is incomplete.

PRISMA Flow diagram should be accredited for on Figure 1´s legend. Second bubble for exclusions should include n=30 for consistency. Figure 1 should be presented in Results. Methods should strictly restrict narrative to the actions taken without providing numbers of findings.

It is recommended that studies listed on Table 1 and following Tables are ordered chronologically, particularly as they should appear ordered on forest plots.

Please review the sentence: ”Several authors have established that morning cortisol levels in serum [50], saliva [51,52] or hair [52] are lower in subjects with chronic musculoskeletal pain or FM [49].” To clarify which studies included FM patients and which did not.

Table 2, explain meaning of TP. Please review that all abbreviations are fully explained on their first appearance. Once abbreviations are introduced the full explanation should not be longer used. Please check the full manuscript for adherence to this rule.

Labeling should be improved all over; for example on Figures 2 and 3, single asterisk stands for p<0.05 and also for sample 1; medians are referred but on the table values are tagged as Means, etc. The values of PTTs and PPTs should be clearly indicated on the Figures/Tables. The legend should indicate the label of the dotted lines. Symbols on the legends do not appear on Figures, for example “‡”.

Authors should explain to readers the reasons of presenting the data of Ref. 35, as labels for Samples 1-4.

Although a graph is included on what authors label as Figure 2 and Figure 3, they appear as Tables rather than Figures.

Figure 2. add the term “levels” following “cortisol” on the legend. The text for this Figure indicates that of the 10 studies listed on Table 2, only 8 examined cortisol effects. Please clarify to readers the reasons to exclude studies 32 and 36.

In the following sentence: “One of the possible reasons for this is that adherence to treatment in patients with FM is not always the most adequate, determining that only a limited number of randomized clinical trials have addressed this issue” needs to be reviewed. How can the authors conclude that FM patients do not adhere to treatments? Are the authors saying that there are randomized clinical trials addressing the lack of adherence of FM patients to treatment? It seems quite confusing.

Line 36, The sentence: “..but criteria reflecting the present understanding of this disease that could help clinicians and researchers are lacking” fails to clearly indicate how the criteria mentioned would help clinicians and researchers

Line 55, ref. 13 should be between brackets

Line 106, term “disagreements” should appear in singular and followed by the concordant verb

Line 114, explain what (332) refers to, and/or provide a reference

Line 122, remove “28”

Starting on line 128 review the English

Please review tables format to fit case sizes, references format using brackets etc. Indicate the reasons for bolding scores number 3 in the heading. Since none report conflict of interest, it can be annotated in the Table heading as a general feature, eliminating the column

Line 139, ref. 29 should appear in brackets

Lines 154-155 “392 subjects (all women), of whom 212 were allocated to an FM group (all diagnosed with FM) and 170 to a control group.” Please note that 212 + 170 does not equal 392.

Line 174, the meaning of the following sentence is not clear: “However, this effect was ruled out as zero was included.”

Line 230, please adjust the sentence: “observational assessment of results but lacked any form of intervention..” as obviously, the definition of observational study means lack of intervention.

Review spelling throughout the manuscript, for example on Table 2  “an pain perception” should read: and pain perception

Reviewer 2 Report

This is interesting, in an attempt to correlate cortisol production and pain levels. They explained well why only 12 articles in the end were cross referenced. 

paragraph1 1 p 9- how does exercise tie into cortisol levels?.. like they say in last paragraph on that page/

Perhaps in conclusions, a simple statement of how cortisol level tied to pain level. Could this possible lead to treatment options in the fibromyalgia patient?

Overall, very well done.

Round 2

Reviewer 1 Report

Extensive editing of the English language is required in both the response to review comments and the main text of the manuscript.

Detailed responses should clearly indicate what are the changes introduced. Track changes and correct line numbers in the response sheet are recommended.

A question was left unanswered.

Round 3

Reviewer 1 Report

The authors should rephrase the following 2 sentences in the paragraph preceding Conclusions:

  1. ”The low number of studies means we were unable to provide a good estimate of the overall effect on cortisoland pain.” Replace by: We were unable to provide a good estimate of the overall effect on cortisol and pain from the low number of studies...

  1. ”adherence to treatment in patients with FM is not always the most adequate...” Adequacy to adherance is not objectively defined/measured by the authors. In addition, arbitrary judgement on patient´s behaviour should be avoided.

Author Response

  1. ”The low number of studies means we were unable to provide a good estimate of the overall effect on cortisoland pain.” Replace by: We were unable to provide a good estimate of the overall effect on cortisol and pain from the low number of studies...

Response: We modified the sentence as suggested

2. ”adherence to treatment in patients with FM is not always the most adequate...” Adequacy to adherance is not objectively defined/measured by the authors. In addition, arbitrary judgement on patient´s behaviour should be avoided.

Response: We added the next sentence "The reported lack of adherence to treatment in patients with FM results in a low number of published randomized clinical trials with an even lower number of patients in them"
